# Components Interaction of Cotton Stalk under Low-Temperature Hydrothermal Conversion: A Bio-Oil Pyrolysis Behavior Perspective Analysis

**DOI:** 10.3390/polym14204307

**Published:** 2022-10-13

**Authors:** Xiao Yang, Naihao Chen, Shengbo Ge, Yequan Sheng, Kun Yang, Pengmusen Lin, Xuqiang Guo, Su Shiung Lam, Hui Ming, Libo Zhang

**Affiliations:** 1State Key Laboratory of Heavy Oil Processing, College of Engineering, China University of Petroleum-Beijing at Karamay, Karamay 834000, China; 2Jiangsu Co-Innovation Center of Efficient Processing and Utilization of Forest Resources, International Innovation Center for Forest Chemicals and Materials, College of Materials Science and Engineering, Nanjing Forestry University, Nanjing 210037, China; 3Higher Institution Centre of Excellence (HICoE), Institute of Tropical Aquaculture and Fisheries (AKUATROP), Universiti Malaysia Terengganu, Kuala Nerus 21030, Malaysia

**Keywords:** bio-oil, cotton stalk, hydro-thermal liquefaction, interaction, multi-variate blending

## Abstract

The conversion of agricultural and forestry waste biomass materials into bio-oil by mild hydro-thermal technology has a positive effect on extending the agricultural industry chain and alleviating the world energy crisis. The interaction investigation of biomass components during bio-oil formation can be significant for the efficient conversion of lignocellulose when different raw materials are fed together. In this paper, a bio-oil pyrolysis behavior (thermogravimetric analysis, TG) perspective component interaction investigation of cotton stalks under low-temperature hydro-thermal conversion (220 °C) was studied. Cellulose, hemi-cellulose, lignin, and protein were used as lignocellulose model components, by their simple binary blending and multi-variate blending and combined with thermo-gravimetric analysis and gas chromatography-mass spectrometry (GC-MS) characterization and analysis. The interaction of different model components and real biomass raw material components in the hydro-thermal process was explored. Results showed that the components of hydro-thermal bio-oil from cotton stalks were highly correlated with the interactions between cellulose, hemi-cellulose, lignin, and protein. During the hydro-thermal process, cellulose and hemi-cellulose inhibit each other, which reduces the content of ketones, aldehydes, ethers, and alcohols in bio-oil. Interaction between cellulose and lignin was obvious, which promotes the formation of oligomers, such as ketones, aldehydes, esters, phenols, and aliphatic, while inhibiting the production of aromatic and multi-hybrid compounds. Otherwise, there was no obvious interaction effect between hemi-cellulose and lignin or between lignin and protein. This research will guide the industrialization of lignocellulose, especially the possible co-feed hydro-thermal conversion technology.

## 1. Introduction

Agricultural and forestry wastes are renewable resources, which are widely regarded as the future energy supply’s main sources. At present, biomass energy accounts for approximately 10% of the global energy supply, and there is room for appreciation [1]. The conversion of biomass and renewable agricultural and forestry waste into bio-oil is an efficient conversion method [2]. The bio-oil obtained can be used for bio-fuels for boilers or automobile engines, raw materials for hydro-gen production, main sources of value-added chemicals, bio-fertilizers, carbonaceous materials (coke, activated carbon, and graphite), and adhesives [3]. It has a good development prospect for alleviating the shortage of fuel oil supply [4]. Hydro-thermal liquefaction has been recognized as an efficient and popular technology in many biomass conversion technologies, such as fermentation, transesterification, pyrolysis and gasification, hydro-thermal liquefaction, etc. [5]. Hydro-thermal liquefaction technology (HTL) has the advantages of mild conditions and no drying of raw materials, and it has low energy consumption compared with other biomass conversion technologies, which is an effective way to realize the high-value utilization of biomass resources [6]. Recent years have witnessed a spurt of progress in biomass hydro-thermal conversion to liquid fuels and high-value-added chemical technology. From the source point of view, the interaction among the three main components (cellulose, hemi-cellulose, and lignin) in biomass has attracted more and more attention [7].

This paper mainly explores the interaction between raw material components from the perspective of the pyrolysis behavior of bio-oil, while the current research on the pyrolysis behavior of bio-oil compounds mainly discusses the estimation of the boiling point distribution of bio-oil components by thermo-gravimetry (TG) and gas phase mass spectrometry (GC-MS) [8].

Worasuwannarak et al. [9] studied the pyrolysis behavior of cellulose + xylan (50:50) and lignin + cellulose (50:50) mixed at the temperature range of 100~600 °C and compared them with the calculated theoretical TG curves with thermogravimetric analysis. The results showed that the experimental TG curve decreased faster than the theoretical TG curve in the low-temperature region, but the decrease gradually weakened with the further increase in temperature, indicating that the interaction between lignin-cellulose and xylan-cellulose during pyrolysis is weaker. Wu et al. [10] studied the interaction between cellulose + hemi-cellulose under rapid pyrolysis conditions at different temperatures, mixing ratios, and mixing methods using Py-GC-MS. It was found that the interaction significantly promoted the formation of hemi-cellulose derivatives (hydroxyacetone, acetone, acetic acid, 3-buten-2-ol, and 1-hydroxy-2-butanone) and CO_2_, and slightly inhibited the formation of levoglucan. Chen et al. [11] analyzed the interaction products of cellulose and lignin in the range of 400~700 °C by GC-MS. It was found that the interaction between cellulose and lignin promoted the formation of acid, ketone, and furfural, and the decomposition products of the cellulose sugar unit also promoted the demethoxylation of lignin, which inhibited the formation of phenol and alkyl phenol in the product. Compared with cellulose, the interaction effect between hemi-cellulose and lignin is not significant [12,13]. Zhou et al. [14] researched the process of fast pyrolysis of biomass ingredients (cellulose, hemi-cellulose, and lignin) interaction, and found that the fast pyrolysis of cellulose and lignin mixture raised the product/oil yield of tar by 15 wt. %, while solid carbon yields decreased by 6 wt. %. However, the interaction between xylan and lignin was not significant (yield variation ≤ 5%), which was the same conclusion obtained by Yu et al. [15].

The research on the interaction of the above lignocellulosic components is still dominated by cellulose, hemi-cellulose, and lignin, while less attention is paid to the interaction of the components with less content in real lignocellulosic crops, such as proteins and the three components. Peterson et al. [16] conducted hydro-thermal liquefaction of a glucose + glycine mixture at 250 °C and 10 MPa. They found that glucose and glycine had a strong interaction, namely the Maillard reaction, and then, further found that the carbonyl functional group was the main functional part involved in the reaction. Therefore, they believed that the peptide bond and glycosidic bond could be hydrolyzed in the hydro-thermal medium, and the similar Maillard reactions between mono-saccharides and amino acids may also occur between polymers, such as oligosaccharides, polysaccharides (starch, cellulose, hemi-cellulose, and glycogen), peptides, and proteins. Fan et al. [17] found that the Maillard reaction influenced the yield and quality of bio-oil during HTL of model carbohydrates (lactose and maltose) and proteins, and the highest yield of bio-oil by hydro-thermal conversion of maltose and lysine mixture was 8.9%. It was further proved that disaccharide forms (lactose and maltose) could also interact with lysine. The research results of Mahadevan et al. [18] confirmed that polysaccharides and proteins in polymer form could also interact. They found that cellulose (polysaccharides) interacted with proteins. Glycosamines from the Maillard reaction decomposed into aldehyde alcohols and nitrogen-free polymers by free radical reaction, which increased the amount of bio-oil and combined it with other components to form nitrogen-containing oil products; the interaction between them produced bio-oil with high yield and high nitrogen content. It can be seen from the above studies that there is a significant indigenous interaction between protein components and cellulose and hemi-cellulose. Therefore, the study on the interaction mechanism between lignocellulosic biomass components should not ignore the influence of proteins with less content in hydro-thermal liquefaction.

Our previous work has provided a major summary of the existing studies on the effects of raw materials and their structures in lignocellulose on hydro-thermal bio-oils [19]. Given the influence of raw material structure on hydro-thermal bio-oil, various pre-treatment methods were used to change the structural characteristics of cotton stalk, effectively improve the conversion activity of cotton stalk under low temperatures of hydro-thermal (180–220 °C), and improve the combustion performance and composition of hydro-thermal bio-oil [20]. However, how the interaction between the material composition of hydro-thermal bio-oil affects our current research is still unclear, but the existing binary blend system research is still in the form of pyrolysis; on discussion and analysis, the temperature range is 100~800 °C, although it can confirm the interaction between every single component, describing hydro-thermal liquefaction (HTL) in the process of cellulose. The interaction between hemi-cellulose and lignin is inappropriate. At the same time, HTL is performed at a relatively mild temperature. Compared with the whole temperature range of pyrolysis, the distribution of compounds in the pyrolysis products is not consistent. At this stage, the complete product of the interaction between lignocellulosic components during low-temperature HTL lacks compound distribution.

In order to deeply understand the above problems, low-temperature hydro-thermal liquefaction (220 °C, 4 h) was used to prepare CS-based bio-oil. Under the same hydro-thermal conditions, the blending model lignocellulosic components (namely cellulose, hemi-cellulose, lignin, etc.) were compared with CS bio-oil from the TG and GC-MS characterization analysis. One way, the present work used low temperature hydro-thermal (220 °C), which means low energy consumption and good industrialization prospects. In the other way, we used the same ratio of the three components (cellulose, hemi-cellulose, and lignin) in the real cotton stalk to simulate the hydro-thermal interaction of real lignocellulosic materials. The results may have reference significance for the possible co-liquefaction of different materials to prepare bio-oil for future industrialization. It provides reference and support for parameter optimization of the lignocellulose hydro-thermal liquefaction process.

## 2. Materials and Methods 

### 2.1. Experimental Materials

Cotton stalks were from farmland in Karamay, Xinjiang, China. Before the hydro-thermal experiment, the CS was smashed into powder (CS powder size ≤ 0.074 mm) and dried at 105 °C for 12 h in an air atmosphere in an oven. Reagents used in the experiment: microcrystalline cellulose ((C_6_H_10_O_5_)_n_) was from Beijing Inoka Technology Co, Ltd. (Beijing, China); sodium lignosulfonate (C_20_H_24_Na_2_O_10_S_2_) was from Tokyo Chemical Industry Co., Ltd.; xylan ((C_5_H_8_O_4_)n) was purchased from Adamas Reagent Co., Ltd. (Beijing, China); glycine (C_2_H_5_NO_2_, ≥98.5%) was from Shanghai Aladdin Biochemical Technology Co., Ltd. (Shanghai, China). Ultrapure water (DI water, 18.25 MΩ*cm) used in this study was prepared by the ultrapure water mechanism of WPUP-UV-20 from Sichuan Water Technology Development Co., Ltd. (Chengdu, China), and the polytetrafluoroethylene lining hydro-thermal reactor (CLF-50) was prepared by Shanghai Yushen Instrument Co., Ltd. (Shanghai, China).

### 2.2. Experimental Method

#### 2.2.1. Hydro-Thermal Liquefaction Process

Raw material and ultrapure water were mixed at a solid-liquid ratio of 1:10, the hydro-thermal reactor was heated at a set temperature at 400 r/min, and the reaction was ended after a certain time. Then, cooled the reactor to room temperature and discharged the gas products (not collected); then washed the reactor thoroughly with sufficient amount of dichloromethane; the filtrate and filter residue were transferred to the Brinell funnel for vacuum filtration; the filter residue (biochar) obtained by drying at 105 °C and air atmosphere was weighed and stored to a constant amount, and the filtrate was statically stratified in a separator funnel and got the lower dichloromethane phase to vacuum distillation at 40~45 °C to obtain bio-oil. 

The adopted raw materials include mono-component (cellulose, xylan and lignin), bi-component (cellulose + xylan, [C-X], with a mass ratio of 2:1; cellulose + lignin, [C-L], with a mass ratio of 1:1; xylan + lignin, [X-L], with a mass ratio of 1:1; cellulose + glycine, [C-G], with a mass ratio of 13:1; xylan + glycine, [X-G], with a mass ratio of 7:1; lignin + glycine, [L-G], with a mass ratio of 7:1) and multi-component (cellulose + xylan + lignin, [C-X-L], with a mass ratio of 2:1:1; cellulose + xylan + lignin + glycine, [C-X-L-G], with a mass ratio of 13:7:7:1).

#### 2.2.2. Characterization of Cotton Stalk Raw Materials and Hydro-Thermal Bio-Oil

The composition of bio-oil was determined by gas chromatography-mass spectrometry (GC-MS, Agilent 7890B/5977). Thermogravimetric analysis of hydro-thermal bio-oil was carried out by STA409 comprehensive thermal analyzer of Netzch, Germany. The test atmosphere was nitrogen, and the heating rate was 10 °C/min.

## 3. Results and Discussion

### 3.1. Thermal Gravimetric Analysis of Mono-Component

The TG and DTG curves of hydro-thermal liquefaction bio-oils of cellulose (C), hemi-cellulose (X), and lignin (L) are shown in Figure 1. The glycine-derived hydro-thermal bio-oil under our hydro-thermal conditions (220 °C for 4 h) is so low that it cannot be detected, thus, the glycine TG-DTG investigation is not presented in Figure 1.

The pyrolysis of every mono-component mainly occurs in the temperature range of 150~310 °C, in which mass is greatly reduced, and the highest pyrolysis rate is reached at 214 °C. The pyrolysis and volatilization of low molecular weight polymers and light components mainly occur in this region, and the mass reduction in each mono-component is more than 40%, indicating that low molecular weight polymers accounted for a considerable proportion. In this temperature range, the descending rate is lignin > hemi-cellulose > cellulose. When the temperature reaches 360 °C, the TG and DTG curves of lignin components tend to be flat, while cellulose and hemi-cellulose still have a small pyrolysis peak at 430 °C. Possibly, this is because there are polysaccharides with amorphous structures in cellulose and hemi-cellulose components, and hemi-cellulose has more branched units than cellulose, so more polymers with certain molecular weights can be produced after hydro-thermal liquefaction [21]. When the temperature reaches 430 °C, the curves of cellulose and hemi-cellulose components tend to be flat. The pyrolysis of each component is completed after 600 °C, and the final cellulose, hemi-cellulose, and lignin fractions are 23%, 31%, and 15%, respectively.

### 3.2. Thermal Gravimetric Analysis of Binary Blend

TG and DTG curves of binary blend hydro-thermal bio-oils with different mono-component are shown in Figure 2.

The DTG curves of the cellulose + hemi-cellulose components (C-X) have two obvious weight loss peaks at 230 °C and 430 °C, respectively, which are the same as those of hemi-cellulose (X). This indicates that the weight loss of the mixed samples is mainly caused by hemi-cellulose, the main pyrolysis temperature range of cellulose + hemi-cellulose (C-X) is widened (from 150~310 °C to 100~400 °C), and the weight loss rate is decreased compared with the cellulose DTG curve; and then, the weight loss peak moves to the high-temperature side (the maximum pyrolysis rate peak changed from 210 °C to 230 °C), indicating that the pyrolysis of cellulose mono-component water is inhibited to some extent, which is consistent with the results of Wang et al. [22]. Compared with cellulose, hemi-cellulose can be pyrolyzed in advance at low temperatures, and its liquid state products can be wrapped on the surface of cellulose, which inhibits the volatilization and precipitation of cellulose pyrolysis products, and further improves the temperature limit of cellulose pyrolysis so that the weight loss peak of cellulose extends to the high-temperature side [12]. In addition, most of the volatile components generated during pyrolysis are mostly similar to those of cellulose, which increases the concentration of volatile components in the main pyrolysis temperature range of cellulose, thus, reducing the weight loss rate of cellulose. 

Compared with the mono-component pyrolysis process of cellulose (C) and lignin (X) in Figure 1, the DTG curves of cellulose + lignin (C-L) slowdown in the temperature range of 100~400 °C, but significantly widen the main pyrolysis range of cellulose, indicating that there is a significant indigenous interaction between lignin and cellulose. Wu et al. observed similar results using Py-GC/MS [23]. Due to the pyrolysis process of lignin being earlier than that of cellulose, the initiator group needed for cellulose to break is partly occupied by the pyrolysis products of lignin in the low-temperature pyrolysis interval, resulting in insufficient initiator groups for activated cellulose, and the pyrolysis reaction is hindered [24]. Compared with mono-component hemi-cellulose and lignin, hemi-cellulose + lignin (X-L) can slightly increase the weight loss rate of hemi-cellulose, but the effect is not obvious.

The TG and DTG curves of the hydro-thermally liquefied bio-oil mixed with one component and glycine are shown in Figure 2c,d. The cellulose + glycine (C-G) DTG curve has four obvious weight loss peaks in the temperature range of 100~500 °C. These peaks mainly occur in the pyrolysis at 132~269 °C, and the mass loss is approximately 30%; after 500 °C; the TG and DTG curves tend to be flat, and the pyrolysis is completed after 680 °C. Compared with the cellulose mono-component (single weight loss peak), the cellulose + glycine (C-G) weight loss peak is more diverse, and the weight loss rate greatly slows down, indicating that glycine has a strong interaction with cellulose and produces compounds with different molecular weights. This is similar to the results obtained by Peterson et al. [16], in which cellulose showed a strong interaction reaction with glycine. Current research for wood fiber components and protein is concentrated in the fiber of hemi-cellulose and lignin. Therefore, in this article, through the cellulose, lignin, and pyrolysis behavior of glycine interaction to explore, we supplement the hemi-cellulose and lignin with glycine hydro-thermal liquefaction under the interaction of results. Compared with the mono-component DTG curve of hemi-cellulose, there is a wide and thick maximum weight loss peak in the DTG curve of hemi-cellulose + glycine (X-G), and compared with the DTG curve of hemi-cellulose, the temperature of the weight loss rate peak shifts to the low-temperature side, which indicates that hemi-cellulose can also interact with glycine, but its interaction is weaker than that of cellulose + glycine (C-G). Since hemi-cellulose contains different mono-saccharide units, it can also react with glycine by the Maillard reaction. However, compared with the glucose unit with a single cellulose and a strong reaction activity, the reactive components of various mono-saccharide branched hybrids containing xylose and pectin are less and the branched hybrids are too much. Therefore, the macro performance of the DTG curve is a single and broad weight loss peak [25]. Compared with its mono-component, lignin + glycine (X-G) is still a single weight loss peak. It can still be seen that it is mainly a mono-component lignin DTG curve peak. However, the main pyrolysis temperature range becomes wider, indicating that the interaction between lignin and glycine is not obvious.

### 3.3. Thermal Reanalysis of Multi-Variate Blending

As shown in Figure 3, the TG curves of various components of hydro-thermal bio-oil in multi-component blending are consistent with the real CS, but there are differences mainly in the temperature range of 200~400 °C. The weight loss rate of the actual CS hydro-thermal bio-oil TG curve is lower than the two other components. The differences between each component in the above temperature range can be seen from the DTG diagram. In the temperature range of 200~280 °C, the three components (C-X-L) have a unique and wide-thick weight loss peak. Compared with the DTG curve of binary blending of each mono-component, that is, the macroscopic performance generated from the weight loss peaks of the binary blending components, superposition in this temperature range indicates that the bio-oil compounds interacting with each other between the three components are mainly located in this temperature range.

The DTG curves of the four components (C-X-L-G) show the characteristics of partial cellulose + glycine (C-G) weight loss peaks, where they have three obvious indigenous weight loss peaks (100~181 °C, 200~280 °C, and 300~380 °C), and their temperature range is consistent with that of the cellulose + glycine (C-G) weight loss peaks, but the weight loss peak rate increases in the temperature range of 200~280 °C, and there are some three-component (C-X-L) peaks. Similar to the TG curves obtained by Lu et al. ‘s five-element hydro-thermal liquefaction blending model (cellulose, hemi-cellulose, lignin, lipid, and protein), the mass of the four components (C-X-L-G) all decreased rapidly within the range of 200~380 °C [26]. The compounds situated in the temperature ranges of 100~181 °C and 300~380 °C are mainly produced by the interaction between cellulose and glycine, while the compounds in the temperature range of 200~280 °C are mainly produced by the interaction of each mono-component. There are only two weight loss peaks (100~181 °C and 300~380 °C) in the DTG curve of the CS in the above temperature range, which is consistent with the weight loss peak shape of the four components (C-X-L-G), but the maximum weight loss rate peak is higher than the actual CS and shifts to the low-temperature side. The components of hydro-thermal bio-oil from CS mainly come from the interactions between cellulose, hemi-cellulose, lignin, and protein, and each component is mainly located in the temperature range of 200~400 °C.

### 3.4. Analysis of Bio-Oil Components

The organic components of bio-oil were obtained by GC-MS characterization, and the percentage of peak area of each characterization result in the total area was taken as the percentage of each compound in its bio-oil component. Due to the complexity and diversity of compounds, this paper was divided into 12 categories according to the chemical structure characteristics, including aliphatic hydro-carbons, multiple heterocyclic compounds, aromatic hydro-carbons, phenols, amines, acids, esters, ketones, aldehydes, ethers, alcohols, and others [27].

The GC-MS results of bio-oils from various components are shown in Figure 4. CS is the hydro-thermal bio-oil of CS, in which aromatics and multi-heterocycles account for approximately 25% of its components. Non-aryl compounds, such as aliphatics, acids, aldehydes, ketones, esters, alcohols, and amines, account for approximately 35%, and it is similar to the composition of Zheng et al. [28] obtained from CS bio-oil. Compared with the components of each mono-component, the formation of acids, aldehydes, ketones, ethers, and aliphatic compounds mainly comes from cellulose (C) and hemi-cellulose (X), and the depolymerization, dehydration, and cyclization of various polysaccharides in the main cellulose and hemi-cellulose [15,29]. Various phenols, multi-heterocycles, and aromatic compounds in the bio-oil components of CS mainly come from lignin (L) condensation cyclization and conversion reactions [11].

Combing the TG and GC-MS results for different ratios of model components (cellulose, xylan, lignin, and glycine). Their interaction during low-temperature hydro-thermal conversion is summarized in Figure 5.

When cellulose and lignin (C-L) were co-hydrothermal, the relative area content of non-aryl compounds increased compared with lignin and cellulose, and the relative area content of ketones, aldehydes, esters, and aliphatic groups increased Compared with lignin, aromatic compounds decreased, and phenolic compounds increased, indicating that cellulose and lignin co-hydrothermal could promote each other. The increase in the content of non-aryl compounds was the chain-end mechanism of lignin or lignin derivatives involved in the left-handed glucan of cellulose pyrolysis products and hindered the formation of L-glucan, so that the intermediate at the end of L-glucan could not be decompressed to L-glucan. Finally, low molecular weight-produced compounds, such as esters, aldehydes, ketones, and cyclohexanone, were formed through decomposition [23]. The reason for the decrease in aryl compounds and the increase in phenolic compounds was that cellulose storage enhanced the formation of some lignin derivatives, such as guaiacol, 4-methyl guaiacol, and 4-vinyl guaiacol [12], while cellulose carbohydrate decomposition products could promote the demethoxylation of lignin to inhibit the formation of highly polymerized aromatic compounds accordingly [9]. When cellulose and hemi-cellulose (C-L) were co-hydrothermally heated, there was no significant difference in the types of compounds between cellulose and hemi-cellulose, but the contents of ketones, aldehydes, ethers, and alcohols decreased. This is because the interaction of hemi-cellulose inhibits the formation of ethanol aldehyde and left glucan [30]. Meanwhile, because of the earlier pyrolysis temperature of hemi-cellulose, the hemi-cellulose derivatives produced in advance will cover the cellulose surface, so that the volatile components, such as ethers and alcohols, in cellulose are inhibited relatively easily. The composition of lignin and hemi-cellulose is not different from their mono-components, which is consistent with the results obtained by Yu et al. [15], and without the obvious interaction between them. 

From the co-hydrothermal results of each mono-component and glycine, it showed that among cellulose, hemi-cellulose, and glycine, amines and nitrogen-containing compounds occur the most, while the composition of compounds is mainly concentrated on ketones, aldehydes, acids, and other substances, and the results of cellulose and glycine (C-G) are more obvious, indicating that there is an interaction between cellulose and hemi-cellulose and glycine. The glucose unit in cellulose and hemi-cellulose undergoes carbonyl-ammonia condensation and molecular re-arrangement with glycine, resulting in generating ketones, aldehydes, nitrogen-containing substances, dicarbonyl, and other components with different structures, which makes the components of hydro-thermal bio-oil mainly include ketones, aldehydes, amines, and acids [29]. The composition of lignin and glycine (L-G) is still mainly aromatic compounds, which is basically the same as that of lignin mono-component, indicating that it has no obvious interaction with glycine. 

The co-hydrothermal compositions of three components (C-X-L) and four components (C-X-L-G) in the multi-variate blending are mainly aromatic compounds (aromatic hydro-carbons, phenols) and non-aromatic compounds (aldehydes, ketones, esters, acids, aliphatic hydro-carbons, etc.), but the contents of amines, aldehydes, ketones, and acids in the four components (C-X-L-G) are higher, while it is closer to the components of real CS hydro-thermal bio-oil, indicating that the interaction among the three components in CS hydro-thermal bio-oil is the main source of different types of compounds in CS hydrothermal bio-oil. However, the existence of amino acids will increase the content of amines, aldehydes, ketones, and acids, which makes the multi-component hydro-thermal bio-oil closer to the real hydro-thermal bio-oil composition of CS.

## 4. Conclusions

In this paper, from the perspective of the composition and pyrolysis behavior of hydro-thermal bio-oil, an experimental study on the preparation of hydro-thermal bio-oil under mild hydro-thermal conditions (220 °C, 4 h, 4 g CS, 40 mL of water) was used to investigate the TG and chemical composition of hydro-thermal bio-oil from CS in different single-component, binary- and multi-component blends of CS. The results mainly include: (1) The components of hydro-thermal bio-oil from CS mainly come from the interactions between cellulose, hemi-cellulose, lignin, and protein, and each component is mainly distributed in the temperature range of 200~400 °C, in which there have low molecular weight polymers and volatile light components. (2) Cellulose and hemi-cellulose have mutual inhibition, and their interaction reduces the content of ketones, aldehydes, ethers, alcohols, and other compounds. (3) There is a great interaction between cellulose and lignin, which promotes the formation of oligomers, such as ketones, aldehydes, esters, phenols, and aliphatic, and inhibits the production of aromatic groups and multiple hybrid compounds. (4) There is no obvious interaction between hemi-cellulose and lignin. (5) The interaction between cellulose, hemi-cellulose, and protein components will further aggravate the types of hydro-thermal bio-oil compounds and significantly increase the formation of nitro compounds. The interaction between lignin and protein is not obvious. (6) Although the interaction among cellulose, hemi-cellulose, and lignin is similar to the single component in a compound, the results of the interaction between them are not simple linear superposition but more complex non-linear relationships, which require further mathematical model calculation.

The research content of this paper will deepen the understanding of the hydro-thermal process of lignocellulose raw materials (especially agricultural and forestry wastes) and will have a positive significance and influence on the composition of lignocellulosic biomass interaction compounds under hydro-thermal liquefaction and the construction of wood fiber hydro-thermal conversion.

## Figures and Tables

**Figure 1 polymers-14-04307-f001:**
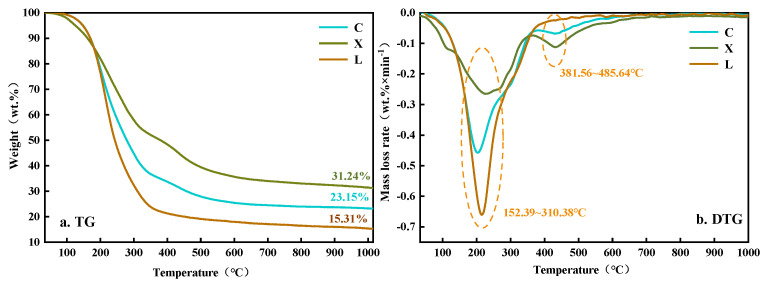
TG−DTG of different single-component hydro-thermal bio-oils.

**Figure 2 polymers-14-04307-f002:**
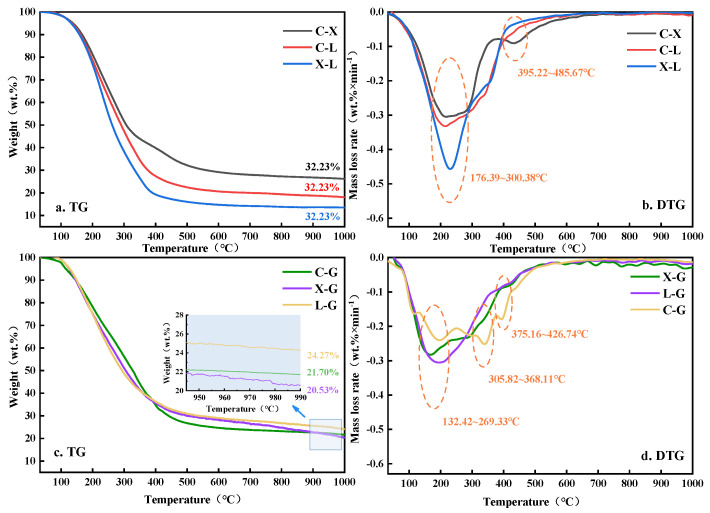
TG−DTG of binary blend hydro-thermal bio-oils with different mono-component.

**Figure 3 polymers-14-04307-f003:**
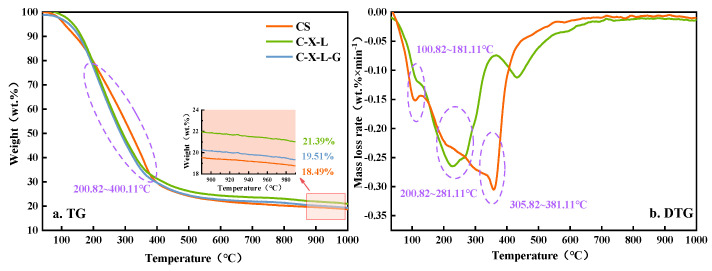
TG−DTG of multi-variate blending hydro-thermal bio-oils in different mono-component.

**Figure 4 polymers-14-04307-f004:**
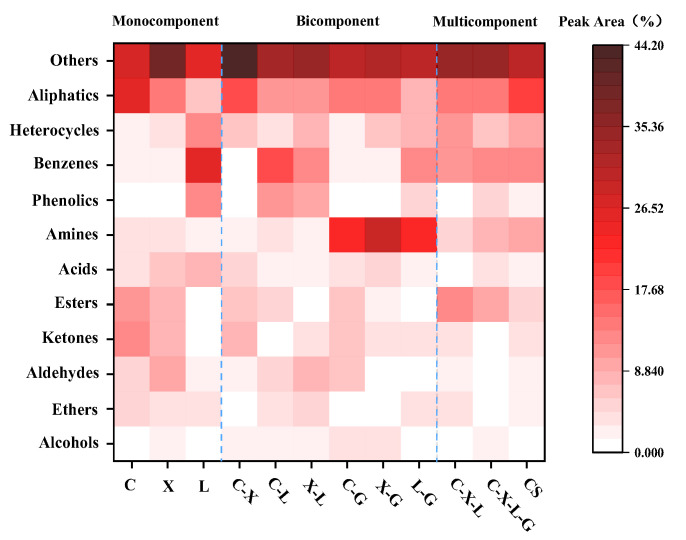
Hydro-thermal bio-oil compounds are distributed in different mono-component, binary blends, and multi-variate blends.

**Figure 5 polymers-14-04307-f005:**
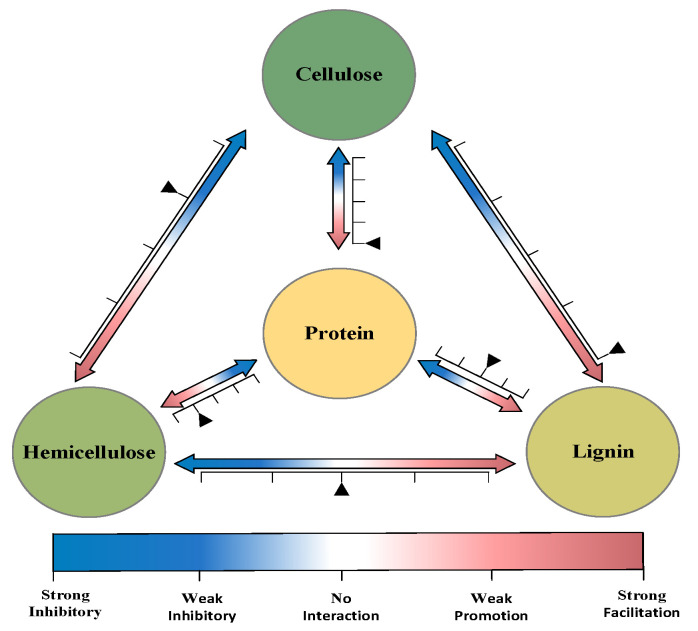
Model components interaction strength schematic during low-temperature hydro-thermal conversion.

## Data Availability

The data supporting the conclusions are included in the main manuscript.

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
