# Peer review of "Components Interaction of Cotton Stalk under Low-Temperature Hydrothermal Conversion: A Bio-Oil Pyrolysis Behavior Perspective Analysis"

_polymers, 2022, doi:10.3390/polym14204307_

Round 1
Reviewer 1 Report
A manuscript contain an original research work, where the authors have presented the results of the investigation of the interaction of different model components and real biomass raw materials components in the hydrothermal process. Cellulose, hemicellulose, lignin and proteins were used as lignocellulose model compounds. Their binary and multicomponent blends were tested with thermogravimetric analysis and chromatography-massspectrometry analysis. I recommend accepting current manuscript however there are some aspect in this manuscript that should be improved:
1. The introduction ends with phrase that the optimization of the parameters of the liquefaction process of wood fibers was carried out. However, all experiments were performed under the same conditions described in experimental part (section 2.2.1). Moreover, the conclusion does not mention any optimization of parameters of the process of liquefaction of wood fibers. It is necessary to explain what the authors mean and, if optimization of process parameters was performed, how it was done. It also states in the introduction that the mechanism of interaction of wood fiber components was proven. However, the subtle mechanism of the chemical interaction of these components has not been established.
2. English requires editing.
Author Response
Thank you for your letter and for the reviewers’ comments concerning our manuscript for publication in Polymers. These comments are all valuable and very helpful for revising and improving our paper, as well as the important guiding significance to our researches. We have studied comments carefully and have made corrections point-to-point which we hope meet with approval and marked with yellow, blue and pink (pink indicates the language polish).

Reviewer 2 Report
Dear Authors and Editors,
I have carefully studied the materials of the manuscript, and below I present my comments and recommendations, which, I hope, will improve the quality of the manuscript.
1) I think that the literature review can be expanded, in particular, by explaining why the authors use only TG-DTG and GC-MS, without FT-IR, elemental composition and other methods that would allow the authors to confirm and more fully explain their results. Additionally, the authors can focus on the relevance of low-temperature hydrothermal liquefaction technology in comparison with other methods of biomass processing.
2) The work actually studied the interaction of microcrystalline cellulose, sodium lignosulfonate, xylan and glycine during the hydrothermal liquefaction process. These components are present in other biomaterials, not only in cotton stalks (CS). Therefore, the significance of the results can be expanded. Otherwise, the authors should explain why the results are limited to CS only. In this regard, the question arises about the correctness of the manuscript title.
3) The last paragraph of the Introduction corresponds with the Materials and Methods section. Instead, authors are recommended to indicate the aim of the work, the uniqueness of the proposed approach and its significance.
4) The size of the smashed CS is important to specify. The powder can be of different sizes.
5) Subsections 2.1 and 2.2 have the same title. Also, the title is missing for subsection 3.1.
6) Why was not glycine investigated by TG-DTG?
7) The authors are recommended to supplement comments from other publications that studied the interaction between hemicellulose + lignin and lignin + glycine, and which, as the authors point out, is not obvious.
8) Authors are recommended to generalize in the Results and Discussion or Conclusions section: Do the interactions of blends comply with the additivity rule?
9) Similar dependences on the interactions of the components under consideration were presented earlier. Therefore, the authors are recommended to add the significance of the results obtained in comparison with previously published works.
10) Some noted technical mistakes:
Line 20. In this paper, A bio-oil pyrolysis behavior (TG) perspective... Replace the preposition A with capital a. TG is not deciphered, although the designation is clear.
Line 24. GC-MS decryption is required.
Line 58. Worasuwannarak et al [8]. studied... Remove point. Same for lines 60, 65, 69, etc.
Line 97. Change dan to the correct word.
Please check the manuscript carefully for similar and other mistakes.
Thanks everyone.
Author Response

(The authors gave the same response as above.)

Round 2
Reviewer 1 Report
Aurtors took into account all the comments of the revieewer. The text has been corrected and supplemented in accordance woth the reviewer' comments. The article can be accepted for publication.
Reviewer 2 Report
Dear Authors and Editor,
I would like to thank the authors for the detailed and consistent explanations on each point of the remarks and the corresponding corrections to the manuscript. I think the manuscript has been improved and could be considered for publication in the journal.